# Resorbable Mg^2+^-Containing Phosphates for Bone Tissue Repair

**DOI:** 10.3390/ma14174857

**Published:** 2021-08-26

**Authors:** Gilyana Kazakova, Tatiana Safronova, Daniil Golubchikov, Olga Shevtsova, Julietta V. Rau

**Affiliations:** 1Department of Materials Science, Lomonosov Moscow State University, Laboratory Building B, 1-73 Leninskiye Gory, Moscow 119991, Russia; dddannn2113@gmail.com; 2Department of Chemistry, Lomonosov Moscow State University, GSP-1, 1-3 Leninskiye Gory, Moscow 119991, Russia; shevtsovaolga7@yandex.ru; 3Istituto di Struttura della Materia (ISM-CNR), Via del Fosso del Cavaliere 100, 00133 Roma, Italy; giulietta.rau@ism.cnr.it; 4Department of Analytical, Physical and Colloid Chemistry, Institute of Pharmacy, Sechenov First Moscow State Medical University, Trubetskaya 8, Build. 2, Moscow 119991, Russia

**Keywords:** whitlockite, calcium magnesium phosphates, struvite, newberrite, bone reconstruction, resorbability, bioactivity, orthopedic applications

## Abstract

Materials based on Mg^2+^-containing phosphates are gaining great relevance in the field of bone tissue repair via regenerative medicine methods. Magnesium ions, together with condensed phosphate ions, play substantial roles in the process of bone remodeling, affecting the early stage of bone regeneration through active participation in the process of osteosynthesis. In this paper we provide a comprehensive overview of the usage of biomaterials based on magnesium phosphate and magnesium calcium phosphate in bone reconstruction. We consider the role of magnesium ions in angiogenesis, which is an important process associated with osteogenesis. Finally, we summarize the biological properties of calcium magnesium phosphates for regeneration of bone.

## 1. Introduction

There is a great interest in bioresorbable materials for tissue engineering in modern surgery. Materials that are similar to native bone tissue are promising. However, in reconstructive surgery and orthopedics, titanium-based metals and their alloys or stainless steel are widely used as orthopedic implants. The main limitations in the use of these metals are due to their undesirable mechanical properties, leading to serious problems of bone remodeling [1,2]. Thus, the absence of degradation of these materials requires a second surgery to remove the implant, and the release of toxic ions as a result of corrosion and microparticles due to material wear can cause inflammatory osteolysis [3,4,5,6]. With long-term use of metal implants and prostheses, there is a high concentration of metal particles in the tissues near the implant, which is the result of the continuous release of metal particles from the implant under mechanical stress [7,8]. Although nondegradable metal implants are generally considered nontoxic, some of their components can contribute to the development of neoplasms [9]. At the moment, cases of the development of osteosarcomas in patients after the implantation of metal endoprostheses have been examined [10]. Thus, there is a need to search for biomaterials for a new generation of implants, which, having the necessary strength characteristics, are biodegradable and do not require repeated surgical interventions for their extraction. The development of such materials will make it possible to shorten the period of restoration of working capacity, as well as to develop the quality of life of the population. In regenerative medicine, biomaterials, including magnesium and its compounds, are relevant and promising for the creation of resorbable biologically active materials in modern implantology [11,12,13,14]. When ingested, magnesium forms chelate-like bonds with many organic substances, thereby ensuring the participation of more than 500 enzymes in metabolic processes—creatine kinase, adenylate cyclase, phosphofructokinase, NAD^+^ kinase, K^+^-Na^+^-ATPase, Ca-ATPase, and many others. Thus, magnesium in the form of coenzymes directly or indirectly participates in the processes of glycolysis, the Krebs cycle, oxidative phosphorylation, protein synthesis, the cycle of urea, glucose and citric acid, metabolism of nucleic acids, lipids, etc. [15]. Therefore, the necessity for secondary surgery for implant removal can be eliminated [16,17,18].

Due to its mechanical properties close to human bone, magnesium allows the elimination of the effects of shielding stress, which contributes to improved biocompatibility of the implant with bone tissue. However, magnesium implants have low corrosion resistance in the body environment, which contains chlorine, where there will later be a premature loss of mechanical properties before the onset of complete recovery of the bone fracture. In connection to this problem, the most developing at present is a regenerative approach aimed at restoring the body’s own bone structures through osteogenesis. It is believed that the body itself can restore lost tissues if certain conditions are created for this. Without external intervention, the cavities become overgrown with fibrous tissue, which has low strength and prevents the transport of nutrients through it; it encapsulates the area of the defect. One of the approaches to prevent the formation of fibrous tissue in the defect area is to fill it with osteoconductive material, which will be a source of phosphate and calcium ions, as the inorganic fraction of human bone consists of hydroxyapatite (HAP: Ca_10_(PO_4_)_6_(OH)_2_) and whitlockite (WH: Ca_18_Mg_2_(HPO_4_)_2_(PO_4_)_12_) [19,20]. Preference is given to materials based on calcium and magnesium phosphates because of their chemical proximity to the mineral component of bone tissue, lack of toxicity, and biocompatibility. Due to the complex bone defects in the body, materials can be filled directly into the defects. These materials include magnesium phosphate cements (MPC), which have several advantages—the ease of use during surgery, and the ability to be resorbed in the body. Recent studies in the field of materials intended for reparative osteogenesis have focused on porous bioceramics, which, on the one hand, are scaffolds for various cells, with biological activity (growth factors, hormones, antibacterial substances, antioxidants etc.) that are released in the environment at a controlled rate. On the other hand, the material must be biocompatible, resorbable, have a system of pores of different modalities (these properties are related to such characteristics as osteoconductivity), and have sufficient strength throughout the period of functioning (implantation and integration into the bone). For the regeneration of bone tissue, ceramics with an ionic type of chemical bond based on calcium phosphates are widely used. However, despite the excellent bioavailability, these substances are not sufficiently resorbable, which does not meet the requirements of a regenerative treatment approach. The use of magnesium phosphates implies a greater solubility of the material in comparison with hydroxyapatite and tricalcium phosphate. Despite the smaller radius of the Mg^2+^ ion in comparison with Ca^2+^, the large hydration enthalpy of the magnesium cation overlaps its contribution to the strengthening of the crystal lattice energy, thereby increasing the phosphate solubility. In addition, possessing special biological functions (suppression of proliferation, osteoclasts and the ability of proliferation and adhesion of osteoblasts), magnesium can shift the balance of bone tissue remodeling toward osteosynthesis.

The increase in the average life expectancy of the population and its growing medical needs have led to the research of new materials for bone tissue regeneration with qualitatively improved properties. Based on the presented data, it can be assumed that biomaterials based on calcium phosphates are widely used; however, biomaterials based on Mg^2+^-containing phosphates can be a good alternative option for use in surgery in the case of bone defects, as they have better properties.

## 2. The Role of Magnesium in the Human Body and Its Inducing Influence on Bone Regeneration

In the human body, magnesium is distributed irregularly: 65% is contained in the inorganic bone matrix, 34% remain in the intracellular space, and 1% is in the extracellular space [21,22]. In cells, magnesium ions occupy the second place after potassium ions and, combining into complexes (80–90%), participate in metabolic processes. They are also distributed to all cellular structures (nucleus, mitochondria, cytoplasmic reticulum, and cytoplasm). The concentration of intracellular magnesium is maintained at a constant level, despite fluctuations in the ion level in the extracellular space. This is due to the relatively limited permeability of the plasma membrane for the cation and the presence of a magnesium transport system [23,24,25,26].

Magnesium is involved in the regulation of the intracellular supply and excretion of calcium through calcium and magnesium-dependent ATPase. It also reduces the release of energy, which is necessary for the penetration of calcium into the cisternae, thereby causing a weakening of the interaction of the contractile proteins actin and myosin in myofibrils and their sliding along one another in the presence of ionized calcium [27]. Magnesium affects the activity of osteoblasts and osteoclasts [28], the concentration of parathyroid hormone, and the active form of vitamin D [29], which are the main regulators of bone homeostasis [15]. Yoshizawa et al. [30] reported that the addition of 10 mM of magnesium in cell cultures of human bone marrow stromal cells (hBMSC) and differentiated osteoblasts enhance mineralization of the extracellular matrix (ECM) by increasing the production of collagen-X and vascular endothelium growth factor (VEGF). Furthermore, they showed (Figure 1a) that magnesium-increased VEGF is co-regulated by hypoxia inducible factor 2a (HIF-2a) in undifferentiated hBMSCs and peroxisome proliferator-activated receptor gamma-coactivator (PGC)-1a in differentiated hBMSCs.

On the one hand, VEGF is essential for the efficient coupling of angiogenesis and osteogenesis during postnatal bone repair [31,32,33,34,35]; it is a major controller of vascular growth. On the other hand, VEGF also inhibits osteoblast differentiation and competes with platelet-derived growth factor (PDGF-BB) for binding with PDGF-Rs (proteins that regulate the proliferation, differentiation, and growth of cells). It deteriorates the function of pericytes, which leads to the formation of immature blood vessels and interrupts the communication of angiogenesis and osteogenesis [36,37,38,39,40]. VEGF may have opposite effects on the physiology of bones under various circumstances (Figure 1b).

Recently, Huang et al. [41] found that an additional 10 mM of magnesium cations activates the canonical signaling pathway Wnt (one of the most important signaling pathways in the stem cell that is necessary for normal differentiation and maintenance of the phenotype). It can also significantly increase the expression of β-catenin and its downstream genes (LEF1 and DKK1), which, in turn, forces hBMSC to differentiate into the direction of the osteoblast lineage and causes an osteogenic effect. In addition, Hamushan et al. [42] reported that the magnesium cations enhance the consolidation in distraction osteogenesis through regulation of the PTCH protein by activating the Hedgehog (Hh) signal transduction pathway, which is an alternative Wnt signaling pathway. Magnesium derived from implants improves the treatment of fractures in rats by promoting the neurological fabrication of CGRP (calcitonin-associated peptide) [43,44]. Xu et al. were the first to demonstrate [45] that the osteogenic effect of magnesium can directly affect bone cells, particularly osteocytes. Extracellular Mg^2+^ via magnesium channels/transport (e.g., TRPM6, TRPM7, and MAGT1) enters bone cells. This leads to a subsequent increase in the level of intracellular cAMP for ATF4-dependent Wnt/β-catenin signaling activation in bone cells (Figure 2). Mg^2+^ deficiency (approximately 0.04–10%) enhances osteoclastogenesis [46,47].

Zhai et al. discovered [48] that magnesium ions suppress the differentiation of osteoclast precursors by inhibiting NF-κB and NFATc1. Mg^2+^ is also involved in osteoimmunological reactions by contributing to the polarization of macrophages toward the M2 phase (which promotes tissue regeneration), in lieu of the M1 phase (which contributes to the inflammatory response) [49,50,51,52]. Generally, magnesium takes a multifunctional role in bone growth and regeneration. It is necessary at all stages of protein molecule synthesis; protein synthesis decreases with the depletion of intracellular Mg^2+^ ions reserves. Magnesium maintains an adequate supply of pyridine and pyrimidine nucleotides, which is necessary for the DNA and RNA synthesis. It acts as a physiological regulator of cell growth [53,54].

The participation of the magnesium ions in human metabolic processes is also determined by physicochemical characteristics. They include a relatively small ionic radius (0.86 Å versus 1.14 Å for Ca^2+^), and high mobility and charge density (Mg^2+^ is usually coordinated by 6-7 H_2_O molecules). It is a strong Lewis acid and, therefore, has a high affinity for strong bases—oxygen-containing ligands, such as water, carbonates, sulfates, and phosphates [55]. The magnesium ion has two hydration shells, which makes the radius of the solvated ion larger than those of other cations (Ca^2+^, Na^+^, and K^+^). This ion has a high hydration energy (≈456 kJ/mol) and a fairly stable coordination number (usually six), implying an octahedral configuration of the first coordination sphere of the ligands. Thus, magnesium, in comparison with the other, most abundant Ca^2+^ ion, wins its competition in many biological processes [24]. In addition, magnesium hydroxide is a weaker base (K_b_ = 2.5 × 10^−3^) in comparison with calcium hydroxide (K_b_ = 4.3 × 10^−2^) [24], which creates a less alkaline environment during the hydrolysis of the corresponding salts; this is important in the case of a large release of these ions in biological fluids to overcome such a phenomenon as alkalosis.

Magnesium is an extremely light metal (1.74 g/cm^3^ density), 1.6 and 4.5 times less than aluminum and steel, respectively [56]. Magnesium’s breaking strength is the best compared to other ceramic biomaterials. The Young’s modulus and compressive yield strength of Mg-based materials are closer to those of natural bone compared to generally used metal implants. Mg^2+^ affects the overall rate of crystallization of amorphous calcium phosphate and the subsequent growth of HAP [57]. The inclusion of magnesium in the hydroxyapatite structure reduces the crystal size and crystal order by replacing calcium with magnesium [58]. Thus, application of magnesium-based implants can develop the strength of the new bone at the implantation site. Mg-substituted hydroxyapatite exhibits high bioactivity and increased osteoconductivity and osteointegration, as an extracellular inorganic matrix [59,60,61]. During the degradation of Mg-based implants, the temporarily accumulated magnesium ions in the implantable bone matrix can be extracted into the circulatory system with no effects on their concentration in the blood serum [58]. The concentration of magnesium in the blood serum (0.65–0.95 mmol/l) remains at normal level for a long time, despite the deficiency of the ions in the tissues. The lack of correlation between the level of serum magnesium and the total content of magnesium in the human body is explained by the fact that ions coming from bones compensate for the decrease in the amount of magnesium [23]. Changes in plasma magnesium levels occur in the case of significant long-term depletion of the ion store. Therefore, Mg is not only a crucial element in the human body, but it also necessitates the evolution of magnesium-based materials, capable of mediating the controlled delivery of magnesium ions.

Around 1938, McBride conducted a large number of tests of the prospective clinical application of magnesium implants. Taking into account the properties of magnesium, he developed a number of methods of work. He also specified that magnesium-based implants are more suitable for use as fixing devices for bone grafts [62]. Besides, Liu et al. [63] noted that the heat-treated magnesium alloy showed improved maintainability as the remaining size of the defect was lower than that of the magnesium alloy without treatment, because of the enhancement in the heat treatment resistance of the magnesium alloy. Brar et al. [64] pointed out that the mechanical properties of magnesium were notably developed when the size of grains of the matrix was reduced with the Sr-addition. To study the effect of elevated extracellular Mg on human osteoclasts, Wu et al. [65] exposed cultures to different concentrations of magnesium. Thus, the degradation effect of the magnesium alloy was simulated. It was found that magnesium chloride initially promoted and then slowed down the development of the proliferation and differentiation of osteoclasts, depending on the concentration, while magnesium extract, apparently, reduced the metabolic activity of osteoclasts. It was shown that magnesium extract at certain concentrations has a positive effect on the formation of osteoblasts, but a suppressive effect on the differentiation of osteoclasts [28].

## 3. Magnesium Phosphate-Based Bone Cements

Magnesium phosphate-based bone cements have a wide range of medical applications as synthetic bone substitutes because of remarkable properties, such as self-aligning ability, high initial strength, biocompatibility, excellent adhesion, and degradability [58,66,67]. Besides, other researchers have emphasized that these materials are promising for bone replacement, in accordance with the degradability and ability to regenerate bone in orthopedic sheep implant models [68]. In addition, an important factor is the installation time [69,70], as the success of a medical intervention depends on it. It has been found that an acceptable installation interval is around 8–15 min [71]. For example, magnesium potassium phosphate cements have a high number of advantages, but they are characterized by a short installation time, which makes them difficult to use [72]. Despite the valuable advantages of magnesium phosphates, these materials characterized by the lack of macroporosity, poor drug release properties, and poor drug delivery properties, which limit their use [73]. However, there are some ways to improve the performance of magnesium phosphate cements.

Zhao et al. [74] aimed to improve the physicochemical and drug release properties, and the biodegradation and biocompatibility of composites through the use of various degrees of crosslinking of gelatin microspheres in bone cements based on magnesium phosphate. In addition, composites of macroporous magnesium phosphate-based bone cements with sustained drug release, built by crosslinking with gelatin microspheres, have demonstrated excellent viability and stimulating effects on the proliferation, osteogenesis differentiation, mineralization capacity, and gene expression (COLI, OPN, and Runx2) of MC3T cells and also showed a strong potential for promoting angiogenesis. To summarize, the addition of gelatin can provide an appropriate environment for cell growth and lead to an enhancement in spread, osteogenesis differentiation, and in the ability to mineralize MC3T3-E1 cells [75,76]. It should also be noted that the gelatinous microspheres accelerated the degradation of macroporous bone cements based on magnesium phosphate. While comparing samples containing different degrees of crosslinking, it was reported that the degradation rate decreases with an increase in the degree of crosslinking [77]. It was also revealed that it is possible to improve the disadvantages of macroporous magnesium phosphate-based bone cements with low porosity and poor drug release properties. According to the described results, it was found that the increase in gelatin amount made the reduction in the pH of composites of macroporous bone cements based on magnesium phosphate more consistent with the physiological environment of humans [78].

It is believed that the most valuable factor affecting the rheological properties of magnesium phosphate cements is their initial hydration rate [79]. It was found that an increase in the interparticle film width increases the space between solid particles and reduces friction between particles [79]. As a result, the yield strength decreases and, therefore, the liquid state between the solid particles is the main factor for the rheological properties of magnesium phosphate cements. It was noted that the more diffused the particles in the system are, the higher the value of the zeta potential is. Thus, the stability of the system increases, but the dispersion can resist aggregation. However, an opposite phenomenon is observed, which consists of the fact that the lower the absolute value of the zeta potential is, the more likely the system should solidify.

Ma et al. [79] reported about the influence of the Mg/P ratio on the rheological properties of magnesium phosphate cements. It was found that the thickness of the water film reduces noticeably with Mg/P ratio, specifying that a higher Mg/P ratio reduces the separation space between solid particles. Based on the presented experimental data, it can be noted that the change in the yield point and plastic viscosity lends itself to an initial decrease and then a gradual increase, depending on the increase in the Mg/P ratio from 2.5:1 to 4.5:1. It should also be noted that the authors revealed that a higher Mg/P ratio reduces the width of the water film between particles and significantly accelerates the initial rate of hydration, which is responsible for the change in the rheological parameters. The Mg/P ratios have an influence on the zeta potential, which changes significantly. The influence of the Mg/P ratio on rheological parameters is hardly interpretable. Thus, electrostatic force cannot be the primary factor that affects the rheological properties of magnesium phosphate cements with different Mg/P ratios.

Shi et al. [80] investigated a way to improve the properties of magnesium phosphate cements. This method consists of adding chondroitin sulfate in different ratios (which enhances the formation of bone nodules and calcium accumulation and promotes osteogenic differentiation of mesenchymal human cells [81]) into the system of magnesium-phosphate cements. The behavior of samples in vitro and in vivo was monitored. It was revealed that the installation time was extended with the increase in the content of chondroitin sulfate for all samples, which is presumably related to the structure of chondroitin sulfate and its effect on the charge of density, which, in turn, affects the hydration reaction. Moreover, there is a decrease in pH value and an increase in the compressive strength of composite cements, depending on the increase in the content of chondroitin sulfate in the samples. In vitro studies have indicated a beneficial effect of the samples on the proliferation, attachment, and differentiation of preosteoblast cells. In vivo studies have shown an increase in bone formation, characterized by the formation of larger and denser bone. To summarize, it can be concluded that with the addition of magnesium phosphate cement to the samples, an improvement in the physicochemical properties of the obtained material can be obtained.

It should be noted that the study of magnesium phosphates covers a wide area of research; however, the above-mentioned articles touch on the most significant aspects related to these materials. Thus, it is possible to highlight the main advantages, disadvantages, and ways of affecting the properties of magnesium phosphate cements, as well as the results of these manipulations. The conclusions are presented in Figure 3.

## 4. Whitlockite Synthesis and Its Bone Remodeling Features

Hydroxyapatite is known as the most thermodynamically stable phase at near-neutral pH values [82]. Whitlockite is a biologically valuable phase in human bones. However, difficulties can appear in the synthesis of this compound, as this phase is thermodynamically stable in a narrow pH values area. Whitlockite exists in biological systems and can be precipitated under acidic conditions, and it can be synthesized in the form of nanoparticles below the boiling point of water. It has a higher stability than that of hydroxyapatite at pH values below 4.2. In addition, it has been suggested that the incorporation of magnesium into whitlockite may be one of the reinforcing factors. The influence of this factor has been preliminarily investigated in other magnesium-doped calcium phosphate systems. For example, the adhesion, proliferation, expression of genes associated with bone mineralization, and the amount of calcium-containing mineral osteoblast deposits, grown on magnesium-doped calcium phosphate compounds, were observed [83,84,85,86].

Jang et al. [87] performed a research that consisted of studying the properties of whitlockite and assessing its biocompatibility. Due to the fact that the theoretical composition of whitlockite is in the area with a stable preference for the precipitation of hydroxyapatite, the synthesis of whitlockite in the ternary system Ca(OH)_2_-Mg(OH)_2_-H_3_PO_4_ is difficult. A suitable method of synthesis was suggested. Whitlockite was synthesized by adding an appropriate amount of H_3_PO_4_ dropwise at a rate of 12.5 mL/min into the solution of Ca(OH)_2_ and Mg(OH)_2_ that was prepared at the suggested ratios of hydroxides. The heat was applied between 60 and 90°. According to the results of the study, pure white nanoparticles were synthesized. They showed excellent biocompatibility, which was comparable to hydroxyapatite. To summarize, the better biocompatibility of whitlockite can be caused by many factors, such as nanostructure, mechanical hardness, and roughness. Studies [87] have also demonstrated that cells grown on a whitlockite granule showed an even better state of proliferation than the level of cell growth on a hydroxyapatite granule.

Moreover, in the process of bone remodeling, osteoclasts create an acidic environment that mobilizes pre-existing minerals with a characteristic phase similar to hydroxyapatite [88,89]. Unlike hydroxyapatite, whitlockite is relatively stable in acidic environments. It is argued that the increased content of whitlockite in adolescent bone allows one to suggest that it can actively participate during the bone remodeling [90,91].

Kim et al. [92] reported that the dynamic phase transformation from whitlockite to hydroxyapatite contributes to the rapid regeneration of bone with a hierarchical nonosseous structure with a higher density. The structural analysis confirmed this fact. In the course of the study, it was shown that whitlockite minerals have the ability to unceasingly release an increased amount of magnesium and phosphate ions compared to hydroxyapatites under physiological conditions. Improved protein adsorption on whitlockite minerals was also confirmed by the in vivo test results, which showed a higher amount of the organic bone formation matrix in whitlockite-based chondroitin sulfate gel implants rather than in hydroxyapatite-based chondroitin sulfate implants. Whitlockite can induce bone regeneration through phase transformation not only quantitatively, but also qualitatively. According to the obtained results, it was concluded that whitlockite minerals stimulate bone regeneration, so they can be used for bone treatment, and the contribution of inorganic minerals in the process of bone remodeling is expressed at low pH.

Difficulties arise in the synthesis of whitlockite, as hydroxyapatite easily precipitates from Ca^2+^ and PO_4_^3−^ containing solution at near-neutral pH. It has been reported that the pure phase of whitlockite nanoparticles can be precipitated in an acidic system with an excessive amount of magnesium ions. The stability of hydroxyapatite decreases under acidic pH and magnesium ions are too small to sustain the crystal structure of hydroxyapatite, thereby preventing its precipitation [93,94,95]. Hydroxyapatite and whitlockite can be transformed into each other via dissolution and re-precipitation processes in the long term by controlling pH [95]. While hydroxyapatite has higher stability under physiological conditions than whitlockite, whitlockite has the superior osteogenic ability [87,92,96]. Even though whitlockite, due to its high solubility, which is greater than that of hydroxyapatite, gradually dissolves under physiological conditions, it can maintain its mass and form for some months both in vitro and in vivo [87,92,96]. Whitlockite bioceramic implants showed a faster resorbability than hydroxyapatite bioceramic implants both in vitro and in vivo [87,92].

Cheng et al. [97] reported that whitlockite promotes the osteogenic activity of cells more than hydroxyapatite. For example, the bone-forming activity of cells was significantly higher when their microenvironment consisted of hydroxyapatite and whitlockite in a 3:1 ratio. As soon as osteoclasts were grown on the surface of bioceramic scaffolds of whitlockite and hydroxyapatite, the resorbed area of scaffolds of whitlockite was twice that of scaffolds of hydroxyapatite [92]. In addition, when a WH-based implant was interposed into a rat calvarial defect model, the resorption of whitlockite was better and quicker than that of hydroxyapatite. Moreover, the resorption rate of synthetic hydroxyapatite was much lower than the regeneration rate of native tissues, probably because of its high crystallinity [98]. To conclude, hydroxyapatite maintains the mechanical stability of the composite hydrogel frameworks, while whitlockite improves the osteogenic capacity of the organic/inorganic hybrid composite frameworks (Figure 4).

## 5. Resorbability of Phosphate-Based Biomaterials with Different Ca/Mg Ratios

The partial substitution of magnesium for calcium cations in hydroxyapatite or tricalcium phosphate (up to 2.4 wt%) is characterized by a reduced degree of crystallinity, large pore size, and certain surface area [99,100,101,102,103]. The appearance of magnesium cations in the structure reduces the parameters of the crystal lattice in accordance with its lower radius of ions (0.065 nm), which leads to the stability of the structure. It also decreases the solubility [101,104]. In this case, the substitution of Mg^2+^ for Ca^2+^ ions in tricalcium phosphate and hydroxyapatite in an amount up to 14 mol.% occurs through the formation of a solid solution [105,106]. An increase in the substitution of magnesium for calcium up to ~20 mol.% leads to the formation of a low-crystallinity phase. A completely amorphous phase occurs in the range 35–50 mol.% [107,108,109,110,111,112,113,114,115,116]. Consequently, TCP and HAP doped with magnesium exhibit increased solubility. However, Gallo et al. [117] studied the resorption behavior of bioceramics based on undoped and Mg-doped β-TCP (1 and 6 mol%, respectively). An alternative to osteoclast culture (pH 4.4) was implemented for 1 h to define the characteristics of the material stimulation for resorption. It was demonstrated for the first time that crystal orientation is a discriminator between grains that resorbed faster and grains that resorbed slower. It is possible to regulate the kinetics of resorption by dosing β-tricalcium phosphate with the ions of interest. Magnesium doping affects the β-TCP lattice parameters and, in addition, stabilizes the β-TCP phase against dissolution. Therefore, the orientations of the crystals, which were predominantly resorbed, changed, which explains the decrease in solubility. In addition, Lee et al. [118] stabilized calcium phosphates, such as brushite (CaHPO_4_)∙2(H_2_O)) and tricalcium phosphate (Ca_3_(PO_4_)_2_), which are thermodynamically unstable under physiological conditions, by replacing the calcium cation with magnesium. The addition of magnesium successfully stabilized brushite in an aqueous solution at pH 7.5 for 12 h at room temperature. The conversion of brushite to apatite usually occurs at elevated pH values. While the Mg content increases, the surface energy of the particle reduces, and thus, the particles become more spherical. Brushite with 14% magnesium substitution still retains a lamellar morphology, but the particles are smaller and thicker. The substitution of up to 50% with magnesium completely transforms it into a spherical nanocrystalline particle (~100 nm). This indicates that the brushite structure becomes poorly crystalline and/or disordered and amorphous in the presence of magnesium. Thus, stabilization of the brushite phase under physiological conditions with the introduction of magnesium opens up a large number of bio-related applications. They require the synthesis of CaP phases under physiological conditions in the presence of signaling molecules, as well as cells. This is especially useful for testing the effectiveness of brushite in delivering nonviral genes. In addition, replacing Ca^2+^ with Mg^2+^ can also stabilize β-tricalcium phosphate at high temperatures (up to 1600 °C) [119,120]. The presence of pyrophosphate ions, due to the trend to form complexes in solution, can also contribute to the formation of amorphous precipitates [121]. The formation of amorphous mixed calcium-magnesium phosphate was also noted during the production of bio-cement by the interaction of calcium phosphate, magnesium carbonate, and phosphoric acid [122].

The excessive magnesium content in solution with Ca^2+^ and PO_4_^3−^ can lead to the precipitation of brushite and whitlockite. Boistelle et al. [123] found that, initially, only the amorphous phase precipitates, and brushite exist at 37 °C in urine or aqueous solutions with comparable Ca^2+^ and Mg^2+^ concentrations. Later, amorphous calcium phosphates are converted to either whitlockite or apatite, depending on the composition of the solution. It has also been shown that magnesium is a potent inhibitor of evolution toward apatite. Cheng et al. [124] observed the homogeneous nucleation of unstable amorphous calcium magnesium phosphate in solutions with concentrations of [Ca] = 3 mM and [PO_4_] ≤ 10 mM at 37 °C and then the transformation into apatite, brushite, and whitlockite (and newberite) depending on the values of the Mg/Ca ratio and the [PO_4_] concentration.

Wu et al. [125] studied the phenomena of bone regeneration of the left femur in white rabbits using a new calcium-magnesium phosphate cement (CMPC). The results showed that CMPC had shorter set-up times and obviously better mechanical properties than those of calcium phosphate (CPC) or magnesium phosphate cements. In addition, CMPC showed a significantly improved degradation compared to CPC in the simulated body fluid. It was shown by cell culture results that CMPC is biocompatible and can support cell proliferation. These results indicate that CMPC satisfies the basic requirements of bone tissue engineering and may also have a noticeable clinical advantage over CPC. It is perspective for use in orthopedic and reconstructive surgery. Klammert et al. [126] reported that a significant enhancement in the properties of brushite cement is achieved through the use of magnesium-substituted β-tricalcium phosphate (general formula Mg_x_Ca_3−x_(PO_4_)_2_ with 0 < x < 3). It has suitable biocompatibility and improves the mechanical properties compared to brushite cement. The introduction of magnesium increases the setting time of the cement from 2 min for a matrix without Mg to 8–11 min for Mg_2.25_Ca_0.75_(PO_4_)_2_ as a reagent. At the same time, the compressive strength of the hardened cement is doubled from 19 MPa to more than 40 MPa after 24 h of wet storage. Magnesium ions slowed down the brushite setting reaction and also formed newberite (MgHPO_4_·3H_2_O) as a second setting product. In other studies [127], it was observed that excessive magnesium oxide residues lead to high pH and poor biocompatibility. Goldberg et al. [128] investigated the influence of [Ca + Mg]/P ratio on the mechanical properties of calcium magnesium phosphates cements. It was also confirmed that the presence of magnesium oxide affects the compressive strength significantly. Besides, it leads to an alkaline reaction that affects cytotoxicity. It is reported that cements with a 1.67 [Ca + Mg]/P ratio demonstrate high compressive strength up to 22 ± 3 MPa. Kowalewicz et al. [129] studied the in vivo degradation, osseointegration, and biocompatibility of three-dimensional (3D) frameworks of CMPC. After 6 weeks of implantation, the Mg225 material based on Ca_0.75_Mg_2.25_(PO_4_)_2_ showed greater osteointegration and volume reduction compared to Mg225d based on Mg225 treated with ammonium hydrogen phosphate (DAHP). DAHP treatment results in struvite deposition. Thus, the size and overall porosity reduce, and the pressure stability increases. All materials showed excellent biocompatibility. They were completely intersected with new bone and the remaining scaffold material was embedded in the native bone. Thus, Mg225 and Mg225d seem to be prospective bone substitutes for a variety of loads that should be investigated further. The efficiency of crystallization inhibitors and modifying additives depends on the reaction conditions [130,131]. There is also another field of interest that represents the production and study of ceramic materials from calcium and magnesium orthophosphates [132,133]. The preparation of ceramics in the quasi-binary system Ca_3_(PO_4_)_2_-Mg_2_P_2_O_7_ based on powders synthesized from calcium and magnesium nitrates and ammonium hydrogen phosphate at various Ca/Mg molar ratios was studied in [134]. The effect of the reaction temperature, concentration, and pH of the initial solutions were considered in [135,136,137,138]. Kitikova et al. showed [139] that the temperature of solutions, the rate of addition of reagents, and the maturation of sediments have an insignificant effect on the characteristics of calcium magnesium phosphates.

## 6. Conclusions

Analysis of the literature showed that, despite the promising use of Mg^2+^-containing biomaterials, several problems impede their clinical use. It follows that the development of new Mg^2+^-containing biomaterials with controlled biodegradation and osteoinduction has great importance for various branches of clinical medicine. It is known that a high proliferative potential of osteoblasts is preserved on smooth matrices, but the osteogenic differentiation of cells is hindered. When creating volumetric implants, the main problems are resistance to mechanical stress and osteointegration with the prevention of the formation of a fibrous capsule around the implant. Randomly organized porosity using, for example, a replica method significantly reduces the strength of the porous ceramic materials against regularly organized porosity using volumetric printing techniques. The use of modern additive technologies makes it possible in a shortest possible time to obtain a three-dimensional object of almost any architecture from a computer model made using computer-aided design systems. The use of this approach in the preparation of resorbable Mg^2+^-containing biomaterials will be suitable for obtaining an osteoconductive macroporous material with sufficient strength that is capable of supporting the growth of newly formed bone into the implant, due to the special architecture of the framework of the interconnected pores. Such a development will make it possible to create implants for the healing of bone tissue defects in the form of an inorganic basis for personalized bone and tissue engineering structures.

## Figures and Tables

**Figure 1 materials-14-04857-f001:**
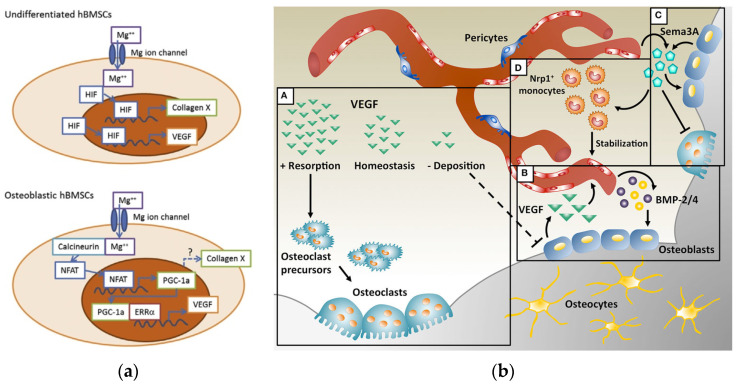
(**a**) Schematic diagram of the putative intracellular signaling cascades upon stimulation of hBMSC with magnesium ions. The addition of a magnesium cation causes an increase in the intracellular Mg ion concentration in undifferentiated BMSCs. The HIFs then migrate to the cell nucleus and induce the production of COL10A1 and VEGF. On the other hand, in differentiated BMSCs, the Mg ion activates the production of PGC-1α (via an unknown transcription factor), which induces the production of VEGF. Adapted with permission from ref. [30]. Copyright 2014, Elsevier. (**b**) Combination of angiogenesis and osteogenesis in intramembranous ossification. (**A**) Physiological levels of VEGF maintain bone homeostasis, whereas too little VEGF interrupts the differentiation of osteoblasts and too much VEGF increases the recruitment of osteoclasts, resulting in bone resorption. (**B**) During bone repair, VEGF is produced by osteoblasts and promotes the migration and proliferation of endothelial cells. In turn, endothelial cells secrete osteogenic factors such as bone morphogenetic protein (BMP)-2 and BMP-4, which support osteoblast differentiation. (**C**) VEGF dose-dependently regulates the expression of the semaphorin class of molecules 3A (Sema3A) in endothelial cells, while Sema3A from various sources inhibits osteoclast differentiation and stimulates bone deposition. (**D**) Sema3A is also responsible for a set of neuropilin-1 (Nrp1^+^)-expressing monocytes that contribute to vascular stabilization. Reprinted from ref. [31].

**Figure 2 materials-14-04857-f002:**
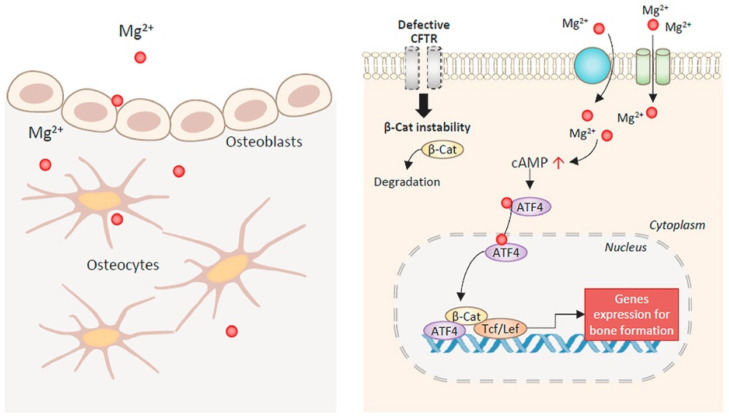
The effect of magnesium on bone formation with CFTR deficiency. Magnesium ions enter bone-forming cells through Mg^2+^ channels or transporters. Mg^2+^ induces the cAMP increase and the activation of transcription factors, ATF4 and β-catenin (β-Cat), rescuing CFTR-deficiency-impaired Wnt/β-catenin signaling to promote bone formation. Reprinted from ref. [45].

**Figure 3 materials-14-04857-f003:**
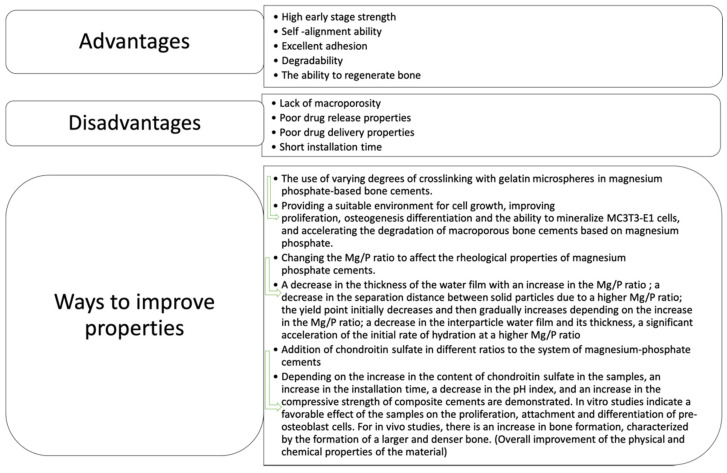
Highlights of the main advantages, disadvantages, and ways of affecting the properties of magnesium phosphate cements, as well as the results of these manipulations.

**Figure 4 materials-14-04857-f004:**
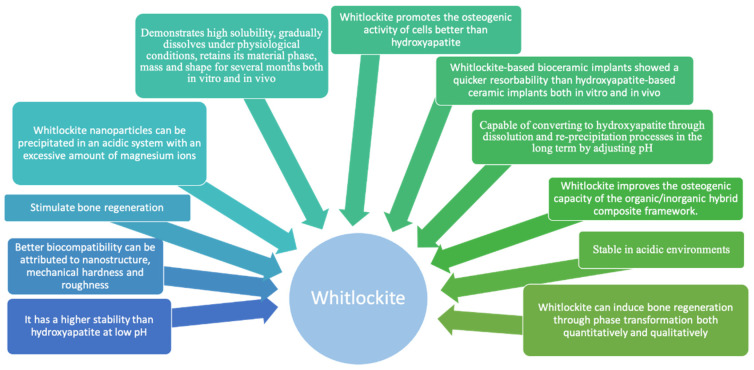
The role of whitlockite-containing biomaterials during implantation.

## Data Availability

All the data is available within the manuscript.

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
