# Peer review of "Resorbable Mg2+-Containing Phosphates for Bone Tissue Repair"

_materials, 2021, doi:10.3390/ma14174857_

Round 1

Reviewer 1 Report

The aim of the article is not stressed.

Line 69, reference 19  is not the research that confirms the whitlockite presence and its significance in human bone. In turn, ref 19 refers to the reports that did not deal with the direct measurements of whitlockite in bones. Following  [S. Tsuboi at el Magnesium Distribution in Human Bone, Calcif Tissue Int (1994) 54:34-37], the concertation of Mg extracted from the bones is around 0.2-0.4 % against the concentration of Ca ~ 20%.

Physiology and biochemistry of  Mg in lines 1378-144 does not relate to the article title key ”… BONE TISSUE REPAIR”

English should be revised. For instance, the order of the words in the sentences, lines 56, 61, etc is not correct. 

Author Response

Dear reviewer,

Thank you for your useful comments which have helped us to improve the quality of the paper! We have re-written a part of the text to answered the comments/suggestions of the reviewers, and the English of the whole text has been edited. The changes we made to your comments in the paper appear in yellow in the revised version. As suggested, we added the correct reference.

 "19. Jackson, S.F.; Randall, J.T. The Fine Structure of Bone. Nature 1956, 178, 798–798, doi:10.1038/178798a0.

20. Scotchford, C.A.; Vickers, M.; Yousuf Ali, S. The Isolation and Characterization of Magnesium Whitlockite Crystals from Human Articular Cartilage. Osteoarthritis and Cartilage 1995, 3, 79–94, doi:10.1016/S1063-4584(05)80041-X."

Reviewer 2 Report

please see an attachment

Author Response

Dear reviewer,

Thank you for your useful comments which have helped us to improve the quality of the paper! The changes we made to your comments in the paper appear in green in the revised version. We added the hypothesis of this paper at the end of the introduction and a bit more information about magnesium phosphate cement in chapter 3. As suggested, two schematic figures in chapters 3 and 4 were added. 

"Based on the presented data, it can be assumed that biomaterials based on calcium phosphates are widely used, however, biomaterials based on Mg2+-containing phosphates can be a good alternative option for use in surgery in case of bone defects, since they have better properties."

"Magnesium phosphate-based bone cements have a wide range of medical applications as synthetic bone substitutes because of remarkable properties, such as self-aligning ability, high initial strength, biocompatibility, excellent adhesion, and degradability [58,66,67]Also, an important factor is the installation time, [69,70], since the success of a medical intervention depends on itIt has been found that an acceptable installation interval is 8 -15 minutes [71]. For example, magnesium potassium phosphate cements have a huge number of advantages, but they are characterized by a short installation time, which makes them difficult to use[72]."

"Yubo Shi et al. [80] investigated a way to improve the properties of magnesium phosphate cements. This method consists in adding chondroitin sulfate in different ratios (which enhances the formation of bone nodules and calcium accumulation and promotes osteogenic differentiation of mesenchymal human cells [81]) into the system of magnesium-phosphate cements. The behavior of samples in vitro and in vivo was monitored. It was revealed that the installation time was extended with an increase in the content of chondroitin sulfate for all samples, which is presumably related to the structure of chondroitin sulfate and its effect on the charge density, which in turn affects the hydration reaction. Moreover, there is a decrease in pH value and an increase in the compressive strength of composite cements, depending on the increase in the content of chondroitin sulfate in the samples. In vitro studies indicated a beneficial effect of the samples on the proliferation, attachment, and differentiation of preosteoblast cells. In vivo studies showed an increase in bone formation, characterized by the formation of larger and denser bone. To summarize, it can be concluded that with the addition of magnesium phosphate cement to the samples, an improvement in the physicochemical properties of the obtained material is observed. It should be noted that the study of magnesium phosphates covers a wide area of research, however, the above-mentioned articles touch on the most significant aspects related to these materials. Thus, it is possible to highlight the main advantages, disadvantages, and ways of affecting the properties of magnesium phosphate cements, as well as the results of these manipulations. The conclusions are presented in Figure 3."
